# Integrated OMICs Approach for the Group 1 Protease Mite-Allergen of House Dust Mite *Dermatophagoides microceras*

**DOI:** 10.3390/ijms23073810

**Published:** 2022-03-30

**Authors:** Rei-Hsing Hu, Chun-Wen Cheng, Chia-Ta Wu, Jiunn-Liang Ko, Ko-Huang Lue, Yu-Fan Liu

**Affiliations:** 1Department of Biomedical Sciences, Chung Shan Medical University, Taichung 40201, Taiwan; hrexing@gmail.com; 2Institute of Medicine, Chung Shan Medical University, No. 110 Sec. 1, Chien-Kuo N. Road, Taichung 40203, Taiwan; cwcheng@csmu.edu.tw (C.-W.C.); allenpig1102@gmail.com (C.-T.W.); jlko@csmc.edu.tw (J.-L.K.); 3Clinical Laboratory, Chung Shan Medical University Hospital, Taichung 40201, Taiwan; 4Department of Emergency Medicine, Changhua Christian Hospital, Changhua 50094, Taiwan; 5Division of Allergy, Department of Pediatrics, Chung Shan Medical University Hospital, Taichung 11819, Taiwan

**Keywords:** house dust mite, *Der m*, cysteine protease, fibrinogen, allergen

## Abstract

House dust mites (HDMs) are one of the most important allergy-causing agents of asthma. In central Taiwan, the prevalence of sensitization to *Dermatophagoides microceras* (*Der m*), a particular mite species of HDMs, is approximately 80% and is related to the IgE crossing reactivity of *Dermatophagoides pteronyssinus* (*Der p*) and *Dermatophagoides farinae* (*Der f*). Integrated OMICs examination was used to identify and characterize the specific group 1 mite-allergic component (*Der m 1*). De novo draft genomic assembly and comparative genome analysis predicted that the full-length *Der m 1* allergen gene is 321 amino acids in silico. Proteomics verified this result, and its recombinant protein production implicated the cysteine protease and α chain of fibrinogen proteolytic activity. In the sensitized mice, pathophysiological features and increased neutrophils accumulation were evident in the lung tissues and BALF with the combination of *Der m 1* and 2 inhalation, respectively. Principal component analysis (PCA) of mice cytokines revealed that the cytokine profiles of the allergen-sensitized mice model with combined *Der m 1* and 2 were similar to those with *Der m 2* alone but differed from those with *Der m 1* alone. Regarding the possible sensitizing roles of *Der m 1* in the cells, the fibrinogen cleavage products (FCPs) derived from combined *Der m 1* and *Der m 2* induced the expression of pro-inflammatory cytokines IL-6 and IL-8 in human bronchial epithelium cells. *Der m 1* biologically functions as a cysteine protease and contributes to the α chain of fibrinogen digestion in vitro. The combination of *Der m 1* and 2 could induce similar cytokines expression patterns to *Der m 2* in mice, and the FCPs derived from *Der m 1* has a synergistic effect with *Der m 2* to induce the expression of pro-inflammatory cytokines in human bronchial epithelium cells.

## 1. Introduction

Allergic asthma is a common airway hyperresponsiveness (AHR) and airway inflammation disease that has a high rate among children and affects the daily life of more than 339 million people [1]. The primary external risk factors are air pollution, outdoor pollens and molds, indoor house dust mites (HDMs), and cockroaches. In tropical regions, HDM allergens are the main risk factors for asthma [2]. In central Taiwan, atopic allergic children experience sensitization to HDMs at a higher rate than other allergens, such as cockroaches, *Aspergillus fumigatus* and pet dander [3].

The most relevant allergen-causing mites species in Taiwan belong to the genera *Dermatophagoidinae*, the major inducers of allergy or asthma, including four species, *Dermatophagoides pteronyssinus* (*D. pteronyssinus*), *Dermatophagoides farina* (*D. farina*), *Dermatophagoides microceras* (*D. microceras*), and *Blomia tropicalis* (*B. tropicalis*) [3]. In an airway inflammation study, the crude HDMs could induce the expression of pro-inflammatory cytokines, such as IL-6 and IL-8, in human epithelium cells [4,5]. IL-8 could recruit neutrophils. Only 50% of asthma cases are associated with eosinophilic inflammation, and other asthma cases are accompanied by increases in neutrophils and IL-8 [6]. The concentrations of IL-8 and neutrophils in the sputum of patients with severe asthma are higher than those in patients with mild and moderate asthma [7]. Neutrophil accumulation may contribute to the development of eosinophilic inflammation in patients with severe and persistent asthma who are insensitive to corticosteroids [8]. HDMs could induce allergy or asthma in children, and a higher proportion of co-sensitization of *Der m* was found compared with other allergens of *Der f*, *Der p*, and *Blo t* in Taiwanese allergic children; however, this species is seldom reported [3].

Fibrinogen is a plasma glycoprotein and an important coagulation factor that is produced in the liver and released into the blood circulation. Its increased levels may be linked to the risk of inflammatory diseases, such as peripheral artery disease, chronic obstructive pulmonary disease (COPD), and asthma [9,10]. Fibrinogen cleavage products (FCPs) could be generated by the allergen source-derived proteases and would trigger allergic responses through Toll-like receptor 4 (TLR4) [11,12]. Landers et al. found that the α chain of fibrinogen is preferentially cleaved by diverse proteinases to generate FCPs, indicating the subsequently innate immune responses through Mac-1 (CD11b/CD18) and TLR4 [13].

*Der p 1* and *Der f 1* are well-known major allergenic components that play a role in allergic airway disease through cellular immune elements and inflammatory cells [14]. Different from *Der f 1* and *Der p 1*, *Der m 1* has a protein sequence of only 29 amino acids. The current knowledge of *Der m* mite allergens is limited, and they are not clearly defined in the WHO/IUIS allergen nomenclature database. In this study, an integrated OMICs approach was used to analyze the *Der m 1* allergen and its recombinant protein production. Whether the *Der m 1* major proteinase-activated immune responses are induced by the generated FCPs was determined. Furthermore, the relationship between proteinase allergen (*Der m 1*) and another major allergen component (*Der m 2*) [15], which is the group 2 allergen of *Dermatophagoides* and thought to be an MD-2-related lipid-binding protein [16,17], was studied in an animal-based model, in addition to the mechanism of FCPs formed by an allergenic proteinase from *Der m 1* in human bronchial epithelium cells.

## 2. Results

### 2.1. Integrated OMICs Analysis of the Potential Der m 1 HDMs Major Allergens from D. microceras

Among the 7,343,792 paired-end cleaned short genomic sequence reads, approximately Q20 1,743,770,754 bases of the HDMs strain *D. microceras* were trimmed by FastQC using the Illumina Solexa^TM^ next-generation sequence (NGS) platform. Draft genome assembly was then drafted using ABySS with a *κ*-mer size of 31 into 5.0327 × 10^5^ putative contigs. The expected size (E-size) of a contig choosing a random base from the assembly is 5016 (Appendix A). The assembled contig (#888199) similarity search against the reference known allergen sequences of *Der f 1* (BAC53948) and *Der p 1* (AAB60215) obtained from the GenBank protein database was performed in accordance with the records on the WHO/IUIS Allergen Nomenclature Sub-Committee website using the BLASTP 2.8.0 + algorithm with expectation values of 4 × 10^−136^ and 3 × 10^−117^, respectively.

Known reference sequences were used to identify the homological sequence with the assembled contigs by GeneWise and obtain the putative *Der m 1* gene structure (Figure 1A). Only 29 residues of *Der m 1* (P16312.1) were found in the database. The candidate *Der m 1* DNA sequence contains six exons and five introns and could be translated into a protein containing 321 amino acids, which is identical to the known compositions of *Der p 1* and *Der f 1* of 84% and 93%, respectively (Figure 2A). Thirteen unique peptides were selected as candidate *Der m 1* proteins from the *Der m* crude protein extracts by proteomic analysis (Figure 1B,C), and the *Der m 1*-unique peptides bands were determined through sequence information by LC-MS/MS spectrums (Figure 2B).

### 2.2. Cysteine Protease Activity of Recombinant Truncated Der m 1 Allergen

The group 1 allergen of *Der m* is an ion-dependent cysteine protease (Figure 2A) and could induce proinflammatory cytokines from respiratory cells [18,19,20]. Hence, the cysteine protease activity of *Der m 1* was studied. The signal peptide cleavage site of *Der m 1* was analyzed by the SignalP 4.1 Server (https://services.healthtech.dtu.dk/service.php?SignalP-4.1 (accessed on 5 March 2022)). The results showed the first 18 amino acids were signal peptides with a cleavage site between Ala18 and Arg19, similar to *Der f 1* and *Der p 1*. The recombinant *Der m 1*, excluding the signal peptide, was constructed, expressed in *Escherichia coli,* and further denatured and refolded (Appendix A). After purification by affinity column, the enzyme activity of *Der m 1* was measured by the fluorescence substrate, Boc-QAR-MCA. The protease activities of the purified *Der m 1* were tested with or without E-64, a cysteine protease inhibitor. The results indicated that the recombinant *Der m 1* has cysteine protease activity (Figure 3A). Enzyme kinetics were investigated using different concentrations of substrates, and the results were displayed in the Lineweaver–Burk plot (Figure 3B). The estimated value of *Km* for recombinant truncated *Der m 1* is 87.01 μM. The cysteine protease activity of the *Der m 1* component can be identified through the protease activity assay and the cysteine protease inhibitor, E-64 (Figure 3A).

### 2.3. Pathophysiological Features of Asthma Using Recombinant Der m 1 Allergen-Sensitized Mice Models

In previous studies, the group 1 allergen of *Dermatophagoides* could induce allergies and asthma [21]. *Der m 1* may also induce asthma in mice. The dose–response curve of direct airway hyperresponsiveness (AHR) was assessed by measuring the changes in lung resistance and compliance elicited by methacholine inhalation [22]. The results of airway resistance and compliance are shown in Figure 4. Increased changes in respiratory system resistance were found after the combined *Der m 1* + *Der m 2* exposure compared with those of the groups of *Der m 1* and *Der m 2* alone in the 20 mg/mL methacholine challenge dose (Figure 4B). *Der m 1* + *Der m 2* significantly caused more severe asthma in mice, and this might be the synergistic effect of the combination of two different groups of allergens.

In addition to the allergen-specific immune response, IgG1/IgE could be a valid indicator for preferential Th2-driven response in Balb/c mice [23], and the specific IgG1/IgE in mice sera for *Der m 1* and *Der m 2* were determined (Figure 4C). *Der m 1*, *Der m 2*, and the combination of both could elevate the IgG1 and IgE levels in mice sera to significantly higher values than that of the control. Although the quantities of IgG1 and IgE for the combination were not significantly different from *Der m 1* and *Der m 2* alone, the averages of the combination in IgG1 and IgE are higher. Therefore, *Der m 1*, *Der m 2*, and their combination could promote serum allergen-specific immune responses and induce lung function change.

### 2.4. Synergistic Effect in the BALF of Allergen-Sensitized Mice Models with Der m 1 + Der m 2 Combined Inhalation

Bronchoalveolar lavage fluid (BALF) can be used to analyze immune responses, such as inflammatory responses caused by allergens, immune mechanisms, and infectious diseases in the pulmonary airways. Lymphocyte-infiltrating cells including eosinophils, neutrophils, lymphocytes, and monocytes in BALF were investigated to evaluate the inflammatory response of *Der m 1*, *Der m 2*, and their combination in the pulmonary airways, and we also investigated the lymphocyte-infiltrating cells. All the infiltrating cells of *Der m 1* or *Der m 2* or their combination were elevated (Figure 5). A statistically significant increase in neutrophils infiltration was found in the group with combination exposure compared to those with exposure to *Der m 1* and *Der m 2* alone. This finding implies that the combination-allergen-challenged mouse model induces a severe inflammation response. Therefore, the lung tissue samples were stained with hematoxylin and eosin to observe the inflammation around the airway. Compared with that of the negative control (normal mice), the airway smooth muscles in the allergen-treated mice were thicker, and the goblet cells suffered from hyperplasia (Figure 6). The effect was more severe for exposure to the combination of *Der m 1 + Der m 2* (Figure 6).

The 111 cytokines profiled in BALF were investigated via principal component analysis (PCA) to further analyze the cytokines in four groups of different allergens. According to the dimension reduction of PCA clustering, the distribution of *Der m* allergens to four groups of sensitized mice was 17.3% and 62.6% of the PC1 and PC2 variances in cytokine levels from the control, respectively. However, the distribution of *Der m 1*-sensitized mice was distinctly different from that of *Der m 2* and their combination (Figure 7A). This finding indicates that the sensitized pathway of *Der m 1* differed from that of *Der m 2*, and the synergistic effect of their combination might be associated with the sensitized pathway of *Der m 2*. In terms of the cytokines contributing to each cluster, IL-6 was increased in the four groups (Figure 7B).

### 2.5. Der m 1 Is a Cysteine Proteinase That Digested the α Chain of Fibrinogen, and the Combination of Fibrinogen Cleavage Products Could Induce the Expression of Pro-Inflammatory Cytokines

The fungal protease or allergen source-derived protease can cleave native fibrinogen into FCPs, which can engage the TLR4 receptor in allergic airway disease [12,13,24]. Regarding the mechanism of *Der m 2* alone and the combination of *Der m 1* + *Der m 2*, we hypothesized that FCPs may play an important role in the synergistic effect of *Der m 1* + *Der m 2*. Firstly, recombinant *Der m 1* was found to cleave the native fibrinogen in vitro and could partially degrade 5.4% to 37.1% of the native fibrinogen α chain under 6–24 h of processing (Figure 8).

FCPs were generated in vitro (*Der m 1*-derived FCPs) from the native fibrinogen through *Der m 1* proteolytic digestion. An epithelial cell line from the human bronchial epithelium, BEAS-2B cells, was used to treat FCPs. The mRNA expression levels of the pro-inflammatory cytokines IL-6 (*p* < 0.001) and IL-8 (*p* < 0.01) were elevated by *Der m 2* alone, *Der m 1*-derived FCPs, or a combination of them both, but not by the native fibrinogen alone or *Der m 1*-derived FCPs alone (Figure 9). Furthermore, the combination of *Der m 1*-derived FCPs and *Der m 2* significantly increased the mRNA expression levels of IL-6 and IL-8 by two- and threefold than those of *Der m 2* alone and *Der m 2* combined with native fibrinogen, respectively (*p* < 0.001, Figure 9). These results indicate that the combination of *Der m 2* and *Der m 1* derived-FCPs synergistically trigger the gene expression of pro-inflammatory cytokines IL-6 and IL-8.

## 3. Discussion

A bioinformatics approach was employed to determine the homologous *Der m 1* from integrated OMICs data by comparing the *Der f*, *Der p,* and assembled genomic contigs of *D. microceras*. The proteolytic and immunomodulation activities for *Der m 1*, *Der m 2* and their combination in causing allergenic hyperresponsiveness and airway inflammation were also studied in the cell-based and sensitized rodents’ models. *Der m 1* could degrade the α chain of the coagulation factor, fibrinogen, but the FCPs from the treatment of *Der f* and *Der p* crude extract proteins had similar results in a previous study [13]. Therefore, *Der m 1* may play one of the important roles of *Der m* allergens, and *Der m* allergens may contribute to allergen-caused asthma in central Taiwan.

Coagulation factors participate in the airway allergic disease, and the FCPs cleaved by proteases would engage innate TLR4 to induce immune responses [12,13]. Allergic proteases of HDMs, *Der p 1,* and *Der f 1* can induce allergy by disrupting tight junctions [25], activating protease-activated receptor (PAR)-1, inactivating PAR-2 [19,26], and degrading and inactivating lung surfactant proteins A (SP-A) and SP-D [27]. In Figure 7B, although the contributions of each component in dimension 2 (PC2) were less than 5%, most of them were reported to be related to protease allergens. In the study of subtilisin, a serine protease from *Bacillus* species, the protease would induce proallergic cytokines, such as thymic stromal lymphopoietin (TSLP), IL-33, and IL-1α, etc., as well as the inflammatory cytokines IL-6, IL-4, and TNF-α [28]. Besides, Florsheim et al. also found that the airways of the subtilisin-sensitized mice had increased lung IL-5-producing type 2 innate lymphoid cells [28], which might be the reason IL-5 would be the major component in the PC2 of PCA. In addition, platelet-derived growth factor-BB (PDGF-BB) is an airway smooth muscle (ASM) mitogen used to promote mice ASM growth, proliferation, and induce ASM hyperplasia, and is associated with airway remodeling, epidermal growth factor (EGF), and vascular endothelial growth factor (VEGF) [29,30]. Therefore, the phenomenon of PDGF-BB and EGF in the major components of PC2 might be due to the protease function of *Der m 1*, such as the fungal protease allergen, to disrupt the interactions between ASM cells and the extracellular matrix [31]. However, while the different cytokine expression profiles of *Der m 1 + Der m 2* and *Der m 1* remain unclear, we suggest the change in cytokine expression profiles might be due to the exposure of the LPS-binding protein, *Der m 2*, or LPS [32].

Fibrinogen is produced in the liver and transported through blood circulation, and epithelium cell culture does not express fibrinogen. Therefore, the effect of FCPs on cell culture could be easily observed. Although the recombinant *Der m 1* could only cleave the α chain of fibrinogen, the α chain is preferentially cleaved by proteases from distinct allergens, and this finding agrees with the study of Lander et al. [13]. The cleavage of β and γ chains is required for bronchial epithelium cells to induce the mRNA expression of IL-6 and IL-8 because the FCPs derived from *Der m 1* cannot evoke their expression.

*Der m 1*, *Der m 2*, and the combination indeed caused mice asthma, and the increased allergen-specific-IgE and -IgG1 values represent the results of the Th2 response (Figure 4), and although the amounts of IgE and IgG1 in the *Der m 1 + Der m 2*-sensitized mice seem to have not significantly increased, its neutrophils are obviously higher (Figure 5). In previous studies, increased sputum neutrophils are highly related to the asthma severity and poor response to the inhaled corticosteroids treatment [33,34]. Therefore, the higher number of neutrophils might be the reason for more severe airway inflammation in *Der m 1 + Der m 2*-sensitized mice (Figure 6).

Although the synergistic effect of combined *Der m 1* and *2* might involve the participation of PAR-1, PAR-2, SP-A, and SP-D caused by *Der m 1*, the PCA patterns of mice cytokines were similar between *Der m 2* only and combined *Der m 1* and *2*. Group 2 allergens of *Dermatophagoides, Der f 2* and *Der p 2,* are MD-2 related lipid-binding proteins [16,17] that bind to LPS and subsequently trigger the TLR4 response. Lin et al. found that PAR-2 is not a key contributor to mucin hypersecretion in human bronchial epithelium cells in vitro [35], and thus might play a less important role in asthma. According to the gene expression profiles (HPA RNA-seq normal tissue) in NCBI, PAR-2 (F2RL1, F2R-like trypsin receptor 1) is extensively expressed in the gastrointestinal tract, gall bladder, kidney, and skin, but TLR4 expression is higher in the lungs than in the skin. Therefore, the synergistic effect of the allergen combination of *Der m 1* and *Der m 2* might have occurred through the FCPs and TLR4 in airway epithelium. In future studies, the roles of other immune cells, such as dendritic cells and macrophages, in the mouse model and asthma must be considered. Moreover, fibrinogen is only produced in the liver. The present study chose human bronchial epithelium cells to examine the innate immune response and elucidate the relationship between FCPs and HDMs allergens. However, the molecular mechanism of the synergistic effect still needs additional experiments through the knockdown or mutation of TLR4.

HDMs allergens can be roughly divided into protease and non-protease allergens. In addition to *Der m 1*, other proteases might participate in the digestion of fibrinogen. Furthermore, a specific fragment of the FCPs could induce inflammation and evoke animal asthma.

The DNA and protein sequences of *Der m 1* were analyzed by NGS data and bioinformatics tools with an integrated OMICs approach and were confirmed to be identical to sequences with the LC-MS/MS spectrum. Recombinant *Der m 1* proteins were further expressed, and their cysteine protease activity was confirmed. The protease activity could proteolytically cleave the α chain of fibrinogen in vitro. Moreover, the cytokine expression patterns of respective *Der m* allergen-sensitized mice were analyzed. The results showed that the synergistic effect of *Der m 1* and *2* was associated with FCPs in human bronchial epithelium cells.

## 4. Materials and Methods

### 4.1. Reagents

The cysteine protease inhibitor E-64, isopropyl-D-thiogalactoside, ovalbumin, *E. coli* lipopolysaccharide (LPS), and Tri reagent^®^ were purchased from Sigma-Aldrich (Burlington, MA, USA).

The mouse anti-*E. coli* LPS antibody and the rabbit anti-mouse HRP antibody were purchased from Abcam (UK).

### 4.2. Next-Generation Sequencing and De Novo Assembly

*Dermatophagoides microceras* was purchased from Thermo Scientific™ ImmunoCAP™ Mite Allergens, and we consigned the samples to Welgene Biotech Co., LTD (Taiwan) for genomic DNA isolation and NGS sequencing by Illumina Solexa™ platform. The paired-end cleaned reads were assembled by ABySS (version 2.1) with 31 k-mer sizes on the Ubuntu 16.04 server. The sequences of allergen proteins of *Dermatophagoides farina* and *Dermatophagoides pteronyssinus* were obtained from the WHO/IUIS Allergen Nomenclature Sub-Committee website (http://www.allergen.org/ (accessed on 5 March 2022)), and they were used for homologous proteins blast by BLASTP 2.8.0+ algorithm. The contigs with similar sequences to the allergen proteins were further compared by GeneWise to identify their cDNA sequences of the putative allergen proteins of *Der m*.

### 4.3. Expression and Purification of Recombinant Proteins

The recombinant proteins were expressed and purified according to a protocol described previously [36]. In detail, the first 18 amino acids were predicted as a signal peptide by SignalP4.1 Server. The cDNA of *Der m 1* without the signal peptide region was synthesized into pET29a.

Recombinant truncated *Der m 1* protein was induced by using 0.1 mM isopropyl-D-thiogalactoside (Sigma-Aldrich, MO, USA) in *E. coli* BL21 (DE3) (ECOS™21; Yeastern Biotech, Taiwan). The *E. coli* harvested by centrifugation at 7000× *g* for 15min was resuspended in the buffer containing 50 mM Tris-HCl (pH 7.5) and 150 mM NaCl and was disrupted by sonication. *Der m 1* was in the inclusion bodies, which is precipitated by centrifugation at 20,000× *g* for 30 min, and the inclusion bodies were denatured by denaturing buffer containing 50 mM Tris-HCl (pH 7.5), 150 mM NaCl, and 8 M urea. *Der m 1* was refolded by rapid buffer change to dramatically decrease the urea concentration before *Der m 1* was purified by Ni-NTA affinity chromatography.

*Der m 2 was* constructed without the first 17 amino acids, which was predicted by the N-terminal signal peptide by the same aforementioned expression system and purification protocol, and further purified with the HiTrap™ Q HP anion exchange chromatography column after affinity chromatography column.

### 4.4. Cysteine Protease Activity and Fibrinogen Cleavage

The protease activity of *Der m 1* was measured using the fluorogenic substrate Boc-QAR-Mca (Peptide Institute, Japan) in 10 mM phosphate buffer, pH 6.0. The reaction contained 2 μM recombinant Der m 1 proteins, 1 mM DTT, and 100 μM Boc-QAR-Mca at 37 °C for the indicated time. For the inhibition reaction, the final concentration of E-64 was 2.5 μM; the *Der m* crude reaction included 10 μg Der m crude extract proteins.

The fibrinogen cleavage by *Der m 1* was performed with 5 mg/mL human fibrinogen (HCI-0150R; Hematologic Technologies, Essex, VT, USA) and 0.1 mg/mL recombinant *Der m 1* at 37 °C for the indicated time. The *Der m* crude extract proteins were used at 0.1 mg/mL in the same condition to treat fibrinogen.

### 4.5. Cell Culture

BEAS-2B, the human bronchus epithelium cell, was cultured in the LHC-9 medium (Gibco™; Thermo Fisher Scientific Inc., Waltham, MA, USA) and the media were refreshed before adding the allergen proteins into the medium for the indicated time according to the protocol in a previous study [4]. After treatment, the media were eliminated and then 1 mL of the Tri reagent^®^ (Sigma-Aldrich, St. Louis, MI, USA) was added at room temperature for 5 min before RNA extraction.

### 4.6. qPCR (Real-Time PCR)

Total RNA was extracted using the Tri reagent^®^ (Sigma-Aldrich, St. Louis, MI, USA). First-strand cDNA was synthesized using a High-Capacity cDNA Reverse Transcription Kit (Applied Biosystems™; Thermo Fisher Scientific Inc., Waltham, MA, USA) and real-time PCR was performed with PowerUp™ SYBR™ Green Master Mix (Applied Biosystems™; Thermo Fisher Scientific Inc., Waltham, MA, USA) by using ABI StepOnePlus (Applied Biosystems™; Thermo Fisher Scientific Inc., Waltham, MA, USA) according to a protocol described in a previous study [37]. The qPCR primers are listed in Appendix A.

### 4.7. Mice

#### 4.7.1. Protocol for Mice Allergen Immunization and Challenge

Balb/c mice were purchased from BioLASCO (BioLASCO Taiwan Co., Ltd., Taiwan). All mice were bred and maintained on a 12-12 light–dark cycle in the animal center of Chung Shan Medical University, and the temperature was maintained at 22–24 °C and the humidity was 55–60%.

The mice sensitization and analysis were performed with reference to previous studies [38,39]; briefly, female Balb/c (6–8 weeks old) were intraperitoneally injected with different respective allergens (50 μg) in the first 3 days, and intra-nasal allergens were administered (50 μg) on days 14, 17, 21, 24, and 27 (Figure 4A).

#### 4.7.2. Airway Hyper-Responsiveness and Bronchoalveolar Lavage Fluid Collection

The allergen-induced-AHR was measured using whole-body barometric plethysmography (Model PLY 3211; Buxco Electronic Inc., Sharon, CT, USA). The pressure change in the chamber during inspiration and expiration with increasing doses of methacholine (Sigma-Aldrich, St. Louis, MI, USA) was calculated and recorded as Penh, which is a dimensionless parameter used to evaluate pulmonary resistance. Each mouse received an initial baseline challenge with saline before the increasing doses of inhaled methacholine (0, 5, 10, and 20 mg/mL). At first, the breathing frequency of the mouse was read and recorded for 3 min, and then was challenged by the spray of nebulized methacholine for 3 min. The pressure waveforms in the box result from the respiratory cycles and contain the peak expiratory pressure (PEP), the peak inspiratory pressure (PIP), and the time of expiratory. The value of Penh was calculated by the following equation: Penh = Pause × peak expiratory pressure (PEP)⁄peak inspiratory pressure (PIP). Finally, the averaged Penh values were reported in percentage in reference to the baseline saline values.

The mice lung lavage through the trachea, with 1 mL of normal saline, was performed for the collection of BALF. The cellularity of BALF was analyzed using a hemocytometer, and the remaining BALF was stored at −80 °C until used in the assay.

#### 4.7.3. The Specific Antibodies in Mice Serum

The OVA-, *Der m 1*-, or *Der m 2*-specific IgE and IgG1 were detected using ELISA. In brief, 96-well microtiter plates were coated with 100 mg/mL OVA, *Der m 1*, or *Der m 2* at 4 °C overnight. Then the plate was washed with PBST before blocking with 3% bovine serum albumin (BSA) at 37 °C for 1 h. After washing the blocking solution, the plate was incubated with 50 μL serial diluted mice sera in 3% BSA at 4 °C overnight. After washing the sera solution, the plate was incubated with optimal diluted horseradish peroxidase (HRP) conjugated anti-mouse isotype-specific antibody (BD Pharmingen™) at 37 °C for 2 h. After washing the HRP conjugated antibody solution, the substrate was added to the plate and the plate was measured for absorbance by the microplate read.

#### 4.7.4. Mouse Cytokine Array

The mouse cytokine array (ARY028, Proteome Profiler Mouse XL Cytokine Array, R&D Systems) was used for the analyses of cytokine profiles of the BALF of mice. According to the manual, in brief, the sample was mixed with the detection antibody at room temperature for 1 h before adding the mixture solution to the array membrane. The membrane was incubated at 4 °C on a shaker overnight. Then the membrane was washed after the incubation, before adding the Horseradish peroxidase-conjugated streptavidin to the membrane at room temperature in a shaker for 30 min. After another wash, the array signals on the membrane were detected by a chemiluminescence image system (MultiGel-21), and the mean pixel densities of the spot were calculated by ImageJ analysis software.

#### 4.7.5. Histograms

After the mice were sacrificed by CO_2_, the lung tissues of mice were immediately soaked and fixed in 10% formaldehyde, and then embedded in paraffin. The paraffin blocks were cut into slices, and the lung tissue sections were stained with hematoxylin and eosin (H&E) to evaluate the pathological changes.

## Figures and Tables

**Figure 1 ijms-23-03810-f001:**
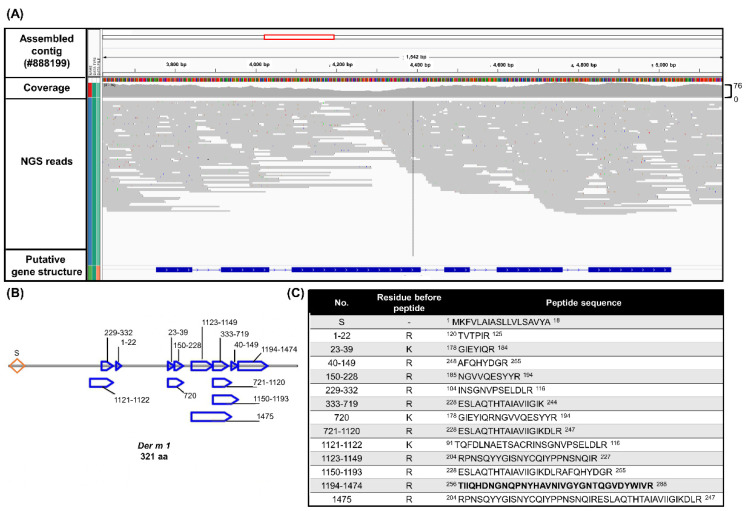
The annotated sequence of the major allergen *Der m 1*. (**A**) The predicted DNA sequence of putative *Der m 1* was shown by Interactive Genomics Viewer (IGV) with the aligned NGS reads as gray lines. The first and second rows show the assembled contig sequence (config #888199; 13,888 bp) and the NGS reads coverage, respectively. The third row shows the sorted NGS reads mapped to the corresponding region mismatching the assembled *Der m* genomic reference indicated by color with alpha transparency proportional to quality. The fourth row shows the predicted exon–intron structure of *Der m 1* using the sim4 package. (**B**) The LC-MS/MS proteomic data showed the specific fragment of the deduced 231-amino-acid sequence of *Der m 1*. The peptide fragments were mapped to the *Der m 1* proteins and the unique peptide sequence of *Der m 1* was shown. For the refused genes track of *Der m 1* gene, the strain is oriented by an arrow while the bar and line indicated the exon and intron region, respectively. (**C**) LC-MS/MS theoretical Mass Spectra approach analysis of *Der m 1*.

**Figure 2 ijms-23-03810-f002:**
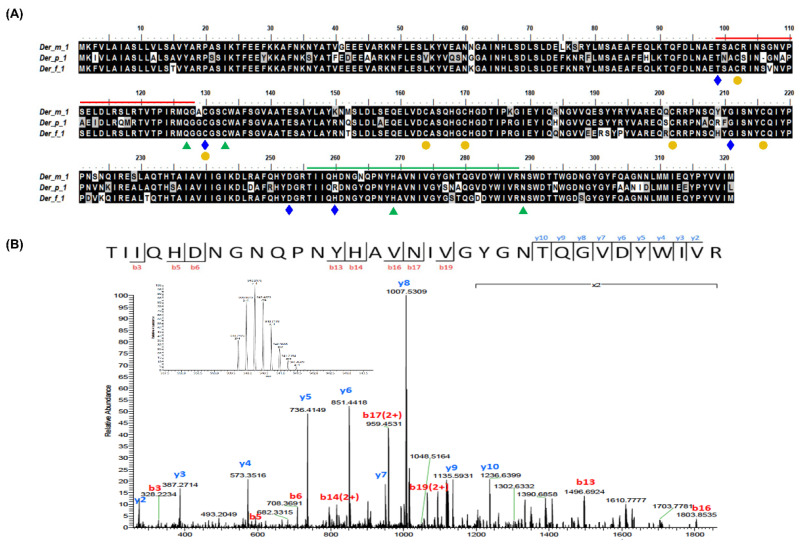
The alignment of putative protein sequence of *Der m 1* with *Der m 1*, *Der f 1*, and *Der p 1* in the Allergen Nomenclature database. (**A**) The open reading frame containing 231 *Der m 1 protein* was aligned with the *Der m 1* sequence (P16312.1, 29 residues, ━) in the Allergen Nomenclature database. The putative *Der m 1* sequence was aligned with *Der f 1* (UniProtKB: P16311.2) and *Der p 1* (P08176.2); the residues with white text on a black background show the identical residues. The five blue diamond symbols (◆) reveal the putative metal ion binding sites for ion-dependent protease activity predicted from NCBI conserved domain database (CDD). The six yellow circle symbols (●) demonstrate the cysteine residues for disulfide bonds forming consistent with protein structures of *Der p 1* and *Der f 1*. Four green triangle symbols (▲) indicate the putative protease active center, green line (━) represents the unique peptide from LC-MS/MS spectra analysis. (**B**) Identification of triply charged precursor ions at *m*/*z* 803.73 of the unique peptide was performed using Orbitrap Elite Hybrid Liquid Chromatographic-tandem Mass Spectrometry (LC-MS/MS) equipped with a Waters nano-Acquity UPLC system. Raw LC-MS/MS data were searched through the Mascot search engine (version 2.2.0, Matrix Science) against putative *Der m* allergen proteins’ primary sequence from NGS analysis.

**Figure 3 ijms-23-03810-f003:**
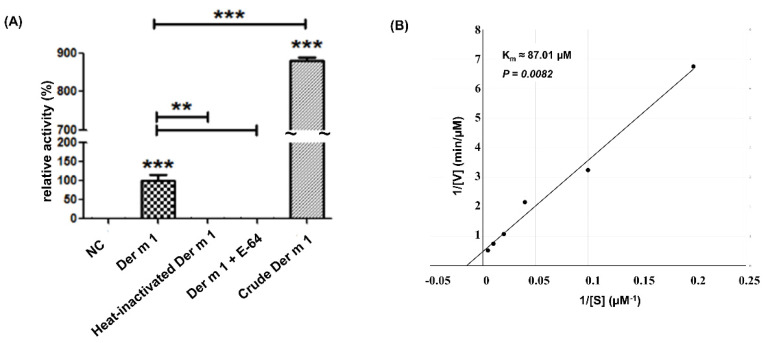
The protease activity assay of the truncated recombinant *Der m 1*. The activity assay used different concentrations of Boc-QAR-MCA as substrate and a final 2 μM recombinant *Der m 1* at 37 °C for 30 min in 96-well plates, determined by ELISA reader. The excitation and emission values are 360 nm and 460 nm, respectively. (**A**) The cysteine protease inhibitor, E-64, indeed decreased the activity of *Der m 1*, as the heat-inactivated *Der m 1* did. (**B**) The Lineweaver–Burk plot showed the Km of recombinant *Der m 1* was 87.01 µM. ** *p* < 0.01, *** *p* < 0.001 compared with NC or *Der m 1* as indicated.

**Figure 4 ijms-23-03810-f004:**
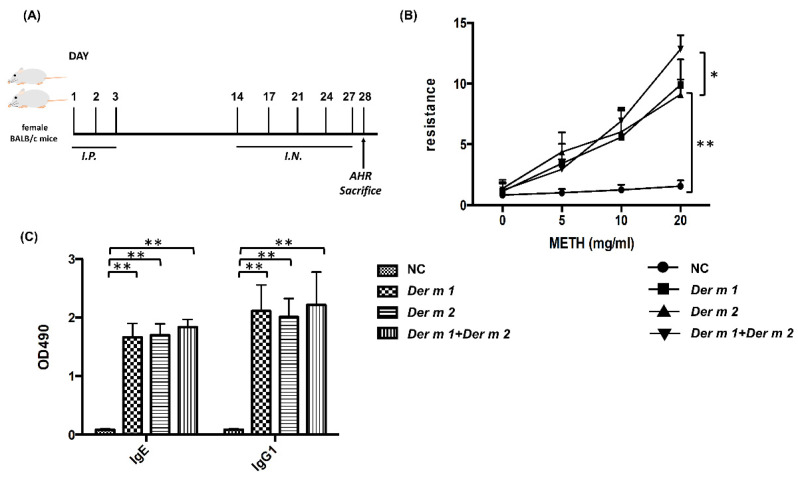
Female BALB/c mice were sensitized/challenged with major recombinant allergens *Der m 1*, *2*, or the combination of *Der m 1* and *Der m 2*. (**A**) In the mice sensitization schedule, the mice were intraperitoneally injected with the allergen or not in the first three days, and were intranasal allergens administration at days 14, 17, 21, 24, and 27, for a total 5 days. After the airway hyperactivity (AHR) determination, the mice were sacrificed on day 28. (**B**) The AHR to methacholine was assessed with the Buxco system. Normal group (NC) is shown as (●), *Der m 1* sensitization group is shown as (■), *Der m 2* sensitization group is shown as (▲), and *Der m 1* + *Der m 2* sensitization group is shown as (▼). (**C**) Sera in allergen-specific IgE and allergen-specific IgG1 concentrations were obtained from the NC group, *Der m 1*, *Der m 2,* and *Der m 1* + *Der m 2* sensitized/challenged mice. ** *p* < 0.01, * *p* < 0.05 compared with NC.

**Figure 5 ijms-23-03810-f005:**
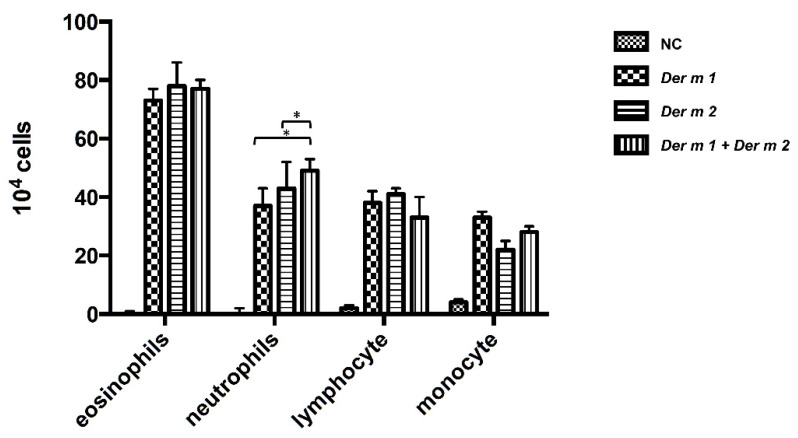
The infiltrating inflammatory cells in the BALF of sensitized mice. All inflammatory cell counts were obtained from the NC group and *Der m 1*, *Der m 2,* or *Der m 1* + *Der m 2* sensitized/challenged mice. The inflammatory cells were counted (×10^4^) from the BALF in millimeters by morphometric evaluations of cytospin preparations. * *p* < 0.05 compared with *Der m 1 + Der m 2*.

**Figure 6 ijms-23-03810-f006:**
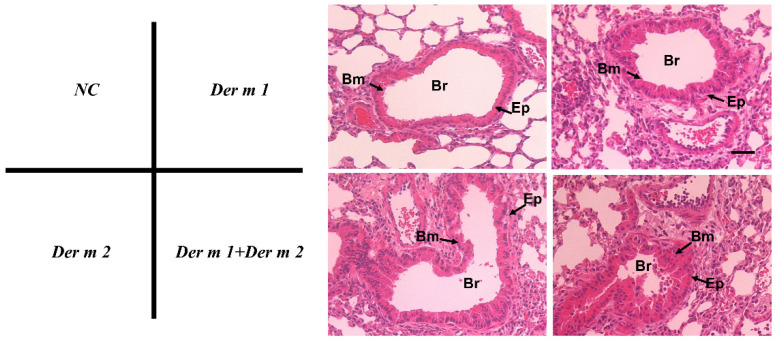
The more severe airway inflammation in *Der m 1* + *Der m 2*-sensitized mice. Lung tissue samples were obtained for histopathological analysis of airway inflammation for the NC group, *Der m 1* sensitized/challenged mice, *Der m 2* sensitized/challenged mice, and a group of *Der m 1 + Der m 2* sensitization mice, respectively. All lung tissue was stained with hematoxylin and eosin to evaluate the inflammation severity and goblet-cell hyperplasia. Br = bronchus, Bm = basement membrane, Ep = epithelium.

**Figure 7 ijms-23-03810-f007:**
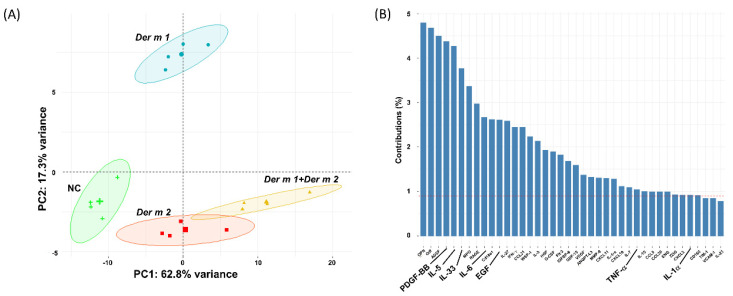
The principal component analysis (PCA) of mouse cytokines protein array profiles of the sensitized mice by different *Der m* major allergens. (**A**) The diagram showed the different groups of four sensitized mice by different allergens, including negative control (NC), *Der m 1*, *Der m 2*, and the combination of *Der m 1* and *Der m 2* (*Der m 1* + *Der m 2*). Each cytokine array contained 111 mouse cytokines (R&D SYSTEMS #ARY028). (**B**) The bar chart showed the principle components in dimension 2 of the PCA results, which is the major difference between *Der m 1* and *Der m 2*.

**Figure 8 ijms-23-03810-f008:**
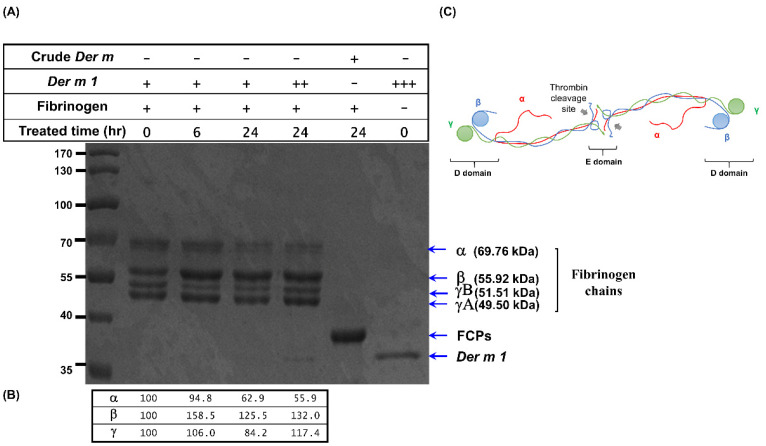
The proteolytic cleavage of fibrinogen by recombinant *Der m 1* protease. (**A**) The fibrinogen (5 mg/mL) was incubated with *Der m 1* (+: 0.1 mg/mL; ++: 0.2 mg/mL; +++: 1.0 mg/mL) and/or crude *Der m* (0.1 mg/mL) at 37 °C for the indicated time (0, 6, and 24 h) and further analyzed by SDS-PAGE. (**B**) The quantity of α, β, and γ chains remaining in the FCPs mixture showing *Der m 1*-mediated degradation of fibrinogen. (**C**) Fibrinogen composed of three different polypeptide chains including α, β, and γ chains. FCPs: Fibrinogen cleavage products.

**Figure 9 ijms-23-03810-f009:**
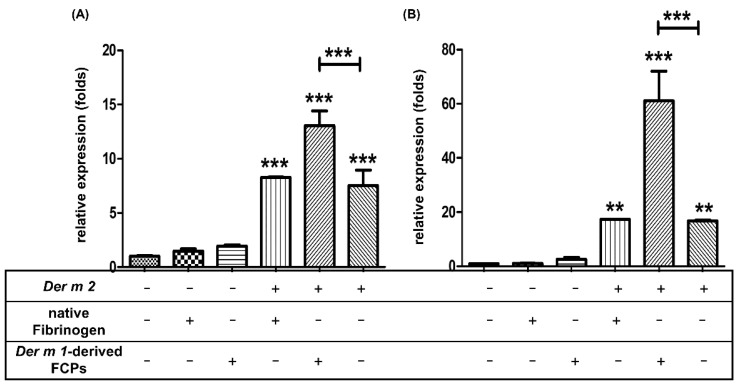
The effects of fibrinogen cleavage products (FCPs) processed by *Der m 1* proteinase in vitro on (**A**) Interleukin-6 (IL-6) and (**B**) IL-8 cytokines mRNA expression induced in human BEAS-2B cells. The native fibrinogen (5 mg/mL) was digested by 0.1 mg/mL *Der m 1* proteinase at 37 °C for 24 h, then the final 1 μg/mL *Der m 1*-derived FCPs were added with or without 1 μg/mL *Der m 2* in the cell culture medium for 6 h treatment. The total RNA from cells was isolated by Tri reagent and reverse transcribed to cDNA before qPCR by specific primers. *** *p* < 0.001, ** *p* < 0.01 compared with control or *Der m 2* as indicated.

## Data Availability

The sequences of *Der m 1* in this study were submitted to DDBJ, and the accession number is LC577890. The detailed biochemical functions of *Der m 2* were confirmed, and its sequence will be submitted once the related research is complete (BDB45822.1).

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
