# Peer review of "Integrated OMICs Approach for the Group 1 Protease Mite-Allergen of House Dust Mite Dermatophagoides microceras"

_ijms, 2022, doi:10.3390/ijms23073810_

Round 1

Reviewer 1 Report

In this manuscript, Hu et al. have employed bioinformatics approach to determine the homologous Der m1 from integrated OMICs data by comparing with similar genes such as Der f and Der p. They, then looked into proteolytic and immunomodulating activities for the Der m1 and m2 in causing the allergenic hyperresponsiveness and airway inflammation in cell-based and sensitized rodents' models. The authors concluded that Der m allergens may contribute to cause allergen and asthma.

The manuscript is well written and covered lot of details. The claims made by authors have been well supported by data. There are couple of minor changes needs to be made which are as follows:

1) In figure-4B, Please include the data points (Sign with what the data represents) by the figure. Something you have shown in figure 4(C). What the round, square and triangle represents by the figure.

2) Please add Y-axis title in figure-9.

Reviewer 2 Report

The aim is stated clearly. The title is informative and relevant. Appropriate and key studies are included.

The study methods are valid and reliable. There are enough details provided in order to replicate the study. The data is presented in an appropriate way. The text in the results add to the data and it is not repetitive. Statistically significant results are clear. It is clear which results are with practical meaning. Results are discussed from different angles and placed into context without being overinterpreted.

However, the discussion fails to provide more specific information on the practical benefits and the translation to the clinical practice, thus; it made the closing statement a bit weaker than it was supposed to be. A carefully tailored conclusion could change the entire quality of the article.

Specific comments on weaknesses of the article and what could be improved:

Major points  - none

Minor points

  1. Please, state the limitations of the study
  2. Could you please discuss the clinical implications of the results
